# Mapping Clinical Questions to the Nursing Interventions Classification: An Evidence-Based Needs Assessment in Emergency and Intensive Care Nursing Practice in South Korea

**DOI:** 10.3390/healthcare13151892

**Published:** 2025-08-02

**Authors:** Jaeyong Yoo

**Affiliations:** Department of Nursing, College of Medicine, Chosun University, Gwangju 61452, Republic of Korea; jaeyongyoo@chosun.ac.kr; Tel.: +82-62-230-6321

**Keywords:** evidence-based nursing practice, Nursing Interventions Classification, clinical questions, intensive care, emergency nursing, evidence needs

## Abstract

**Background/Objectives**: Evidence-based nursing practice (EBNP) is essential in high-acuity settings such as intensive care units (ICUs) and emergency departments (EDs), where nurses are frequently required to make time-critical, high-stakes clinical decisions that directly influence patient safety and outcomes. Despite its recognized importance, the implementation of EBNP remains inconsistent, with frontline nurses often facing barriers to accessing and applying current evidence. **Methods**: This descriptive, cross-sectional study systematically mapped and prioritized clinical questions generated by ICU and ED nurses at a tertiary hospital in South Korea. Using open-ended questionnaires, 204 clinical questions were collected from 112 nurses. Each question was coded and classified according to the Nursing Interventions Classification (NIC) taxonomy (8th edition) through a structured cross-mapping methodology. Inter-rater reliability was assessed using Cohen’s kappa coefficient. **Results**: The majority of clinical questions (56.9%) were mapped to the Physiological: Complex domain, with infection control, ventilator management, and tissue perfusion management identified as the most frequent areas of inquiry. Patient safety was the second most common domain (21.6%). Notably, no clinical questions were mapped to the Family or Community domains, highlighting a gap in holistic and transitional care considerations. The mapping process demonstrated high inter-rater reliability (κ = 0.85, 95% CI: 0.80–0.89). **Conclusions**: Frontline nurses in high-acuity environments predominantly seek evidence related to complex physiological interventions and patient safety, while holistic and community-oriented care remain underrepresented in clinical inquiry. Utilizing the NIC taxonomy for systematic mapping establishes a reliable framework to identify evidence gaps and support targeted interventions in nursing practice. Regular protocol evaluation, alignment of continuing education with empirically identified priorities, and the integration of concise evidence summaries into clinical workflows are recommended to enhance EBNP implementation. Future research should expand to multicenter and interdisciplinary settings, incorporate advanced technologies such as artificial intelligence for automated mapping, and assess the long-term impact of evidence-based interventions on patient outcomes.

## 1. Introduction

In today’s increasingly complex healthcare environment, driven by population aging, the rising burden of chronic diseases, and the rapid evolution of technology-based care, the importance of evidence-based nursing practice (EBNP) is more critical than ever [1]. Among healthcare professionals, nurses in high-acuity settings such as intensive care units (ICUs) and emergency departments (EDs) are expected to make time-sensitive, high-stakes clinical decisions that directly affect patient outcomes [2,3]. These decisions must be both scientifically grounded and contextually responsive.

Despite this demand, the implementation of EBNP in daily clinical practice remains limited [4]. Previous studies indicate that while nurses recognize the value of evidence-based care and express willingness to apply it, they often encounter structural, educational, and institutional barriers to accessing, evaluating, and translating evidence into action [5,6,7]. These challenges are particularly evident in high-acuity environments, where rapid physiological changes and high-risk scenarios frequently arise [8,9]. In such settings, nurses are often the first to recognize early signs of clinical deterioration, interpret real-time physiological monitoring data, and initiate time-critical, protocol-driven nursing interventions when immediate physician presence is not available [2,3,8]. These interventions include triage prioritization, activation of Rapid Response or Code Blue teams, administration of oxygen therapy or intravenous fluids under standing orders, verification and delivery of high-alert medications with strict safety checks, titration of prescribed therapies within authorized parameters, protocol-driven ventilator parameter adjustments within the nursing scope of practice, implementation of infection-control isolation precautions, and execution of Basic Life Support (BLS) or Advanced Cardiovascular Life Support (ACLS) protocols during life-threatening emergencies [8]. Such rapid, high-risk nursing interventions, often performed within seconds to minutes, demand advanced situational awareness, independent clinical judgement, and proficiency in high-stakes decision-making.

To address this practice-evidence gap, the present study is guided by the Iowa Model of EBNP, which provides a structured framework for transforming clinical issues into implementable interventions [10]. This model emphasizes the importance of identifying practice-derived questions, appraising relevant evidence, and implementing context-appropriate strategies to promote high-quality care [11]. Consistent with this logic, the process can be summarized as follows: (1) identification of a problem-focused or knowledge-focused clinical question, (2) retrieval of relevant evidence, (3) critical appraisal and synthesis, (4) application to clinical practice, and (5) outcome evaluation and continuous feedback [1]. Among these steps, the articulation of meaningful clinical questions is recognized as the essential starting point for EBNP [10].

In this study, the Iowa Model was adopted because it provides a trigger-to-action pathway—from problem- or knowledge-focused triggers; through structured question formulation and evidence appraisal; to practice change—that aligns with the aim of translating frontline clinical questions into standardized Nursing Interventions Classification (NIC) taxonomy for an evidence-needs assessment [12]. Alternative EBNP frameworks (e.g., the Johns Hopkins model, the Stetler model, and i-PARIHS) were considered [1]; however, their primary emphasis on research utilization, organizational policy change, or contextual facilitation during later implementation phases was less congruent with the study’s upstream focus on generating and standardizing practice-derived questions for prioritization.

In alignment with the IOWA model, this study captures and classifies clinical questions generated by nurses in ICU and ED settings. Unlike prior research that utilized structured checklists or predetermined items, this study prioritizes open-ended, narrative responses to reflect the authentic, practice-based concerns of frontline nurses. To systematically organize these responses, the study employs the NIC, a standardized nursing taxonomy, to promote consistency, interoperability, and transparency in documenting nursing care [12]. The NIC categorizes interventions into hierarchical domains, classes, and activities and forms a core component of the NANDA–NIC–NOC (NNN) linkage system.

Prior studies have demonstrated the utility of NIC in transforming narrative clinical records and experiential knowledge into a structured format, thereby enhancing communication, transparency, and quality in nursing care [13,14,15]. In stroke care, for example, the cross-mapping of care priorities to NIC interventions has validated the framework’s effectiveness in aligning real-world clinical needs with evidence-based taxonomy [16]. However, few studies have applied NIC to the spontaneous clinical questions generated by nurses themselves—particularly in high-acuity environments such as ICUs and EDs.

Therefore, this study aims to map nurses’ practice-based clinical questions to NIC domains, classes, and interventions; quantify their perceived evidence needs using frequency distribution; and identify high-priority areas for educational planning using Pareto analysis. This approach supports a consumer-centered model of EBNP and facilitates the development of targeted, high-impact educational and clinical strategies. Ultimately, it positions clinical questioning as a meaningful entry point for evidence translation and NIC taxonomy as a bridge between frontline nursing practice and standardized, evidence-informed care.

## 2. Materials and Methods

### 2.1. Study Design

This study employed a descriptive, cross-sectional design using a structured cross-mapping methodology to identify and classify clinical questions posed by nurses into standardized nursing interventions based on the NIC 8th edition. The purpose was to visualize and prioritize evidence needs as perceived by frontline nurses in high-acuity settings and to inform the development of educational and clinical strategies rooted in standardized nursing language. This study also served as a foundational needs assessment for a virtual reality-based simulation education intervention for critical care nurses, which has since been published [17].

### 2.2. Study Participants and Setting

The study was conducted at Chosun University Hospital, a university-affiliated tertiary care hospital located in Gwangju, South Korea, with a total capacity of 849 beds. A total of 112 registered nurses from four high-acuity departments participated: Emergency Department (ED; n = 34), Medical Intensive Care Unit (MICU; n = 25), Neurological Intensive Care Unit (NICU; n = 26), and Emergency Intensive Care Unit (EICU; n = 27). Inclusion criteria were (1) full-time employment as a staff nurse in one of the four departments, (2) a minimum of six months’ experience in the current unit, and (3) voluntary consent to participate. Nurses in administrative or non-direct care roles were excluded.

The ≥6-month threshold was adopted to ensure completion of unit orientation and sufficient adaptation for independent, competent practice in high-intensity environments. This criterion is supported by Duchscher’s Transition Theory—Doing (0–3 months); Being (4–6 months); and Knowing (≥6 months) [18]—as well as Korean empirical evidence indicating that approximately six months are required to provide independent; safe care; engage in interprofessional communication; and respond effectively to emergencies [19,20,21]. To reduce onboarding-related variability and enhance the validity of eliciting practice-grounded evidence needs for nursing interventions, nurses with <6 months in the current unit were excluded.

### 2.3. Data Collection Procedures

Data were collected between May and June 2023. Following institutional approval, a researcher contacted the hospital’s nursing administrator to explain the study objectives. Unit managers were briefed, and sealed envelopes containing survey packets were distributed to each ward by members of the hospital’s nursing education team—who were not involved in the research. Each packet included instructions, a self-administered questionnaire, and a written consent form with a self-generated participant code to ensure anonymity. Completed surveys were returned voluntarily to designated collection boxes placed in each ward. Participation was anonymous and voluntary, and the submission of a completed questionnaire was considered informed consent.

### 2.4. Data Source and Mapping Procedure

The NIC taxonomy, developed by the University of Iowa College of Nursing, is a standardized, research-based taxonomy used to document and classify nursing treatments performed by nurses across various clinical settings [12]. The 8th Edition of NIC includes 614 interventions, systematically categorized into seven domains and 30 thematic classes [12]. Each intervention is uniquely coded and defined with a comprehensive list of associated nursing activities. This hierarchical structure—Domain → Class → Intervention—enables consistent representation; communication; and analysis of nursing care [16]. In this study, this classification framework served as the standard reference for mapping clinical questions to structured nursing interventions, thereby ensuring semantic consistency and facilitating reliable quantitative analysis. All 614 interventions were considered a priori for coding, without restriction to emergency- or critical-care subsets.

The questionnaire asked participants to respond to the following prompt: “If you have a clinical question in the care of the patient in the nursing field, and if you would like to have scientific evidence for that question, please describe it freely.”

Participants were encouraged to express their clinical question either by:(1)Using the PICO format (Patient/Problem–Intervention–Comparison–Outcome), or(2)Writing freely in narrative form, based on their real-world clinical experience.

All responses were transcribed and anonymized to ensure confidentiality. Using a cross-mapping approach adapted from Nonnenmacher et al. [16], each response was examined to identify core nursing actions or intervention intents. These were then systematically mapped to the most appropriate NIC intervention code and categorized by domain and classes, referencing the NIC 8th Edition [12]. To enhance methodological transparency and illustrate the validity of the mapping process, representative examples of how narrative clinical questions were systematically mapped to standardized NIC interventions are presented in Table 1. The mapping was independently conducted by two trained researchers, and discrepancies were resolved by consensus. Inter-rater reliability was evaluated using Cohen’s kappa coefficient to ensure methodological rigor. This process established a standardized coding framework for subsequent analysis.

### 2.5. Data Analysis

Following NIC mapping, the categorized responses were analyzed using descriptive statistics and visual analytics. Frequencies and percentages were calculated for each NIC domain, class, and intervention using IBM SPSS Statistics version 29.0. To determine the priorities of evidence needs, a Pareto analysis based on the 80/20 principle [22] was conducted. Nursing interventions were ranked by frequency, and cumulative proportions were calculated to identify high-priority interventions that accounted for the majority of total responses. The results were visualized using Pareto charts and provided data-driven insights for the development of educational content and the improvement of clinical practice strategies.

### 2.6. Ethical Consideration

This study received ethical approval from the Institutional Review Board of Chosun University (Approval No. 2-1041055-AB-N-01-2023-18) in accordance with the Declaration of Helsinki. Institutional support was obtained from the nursing department of the participating hospital. All participants provided written informed consent after receiving a full explanation of the study and were informed of their right to withdraw at any time without penalty. Confidentiality and anonymity were strictly maintained, and all data were securely stored for research purposes only.

## 3. Results

A total of 204 clinical questions were collected from 112 nurses working in high-acuity departments, including the ED, MICU, NICU, and EICU. The participants had a mean age of 32.2 years (SD = 8.4), and the majority were female (95.5%). The average duration of clinical experience was 8.7 years (SD = 2.9). These open-ended responses were analyzed using a structured cross-mapping methodology and classified into standardized NIC interventions, domains, and classes according to the 8th Edition of the NIC taxonomy [12]. The coding of clinical questions to NIC interventions demonstrated high inter-rater reliability, with a Cohen’s kappa of 0.85 (95% CI: 0.80–0.89), suggesting almost perfect agreement between the two independent coders [23].

### 3.1. Mapping of Clinical Questions to NIC Interventions

Table 1 presents selected examples of how clinical questions were systematically mapped to NIC interventions. These cases illustrate how frontline clinical inquiries—ranging from oxygen delivery methods to safe medication practices—were translated into discrete nursing actions and linked to NIC interventions. For instance, a question on the effectiveness of different oxygen devices was mapped to “Oxygen Therapy (3320)” under the domain Physiological: Complex, while an inquiry regarding post-antibiotic skin test evaluation aligned with “Allergy Management (6410)” in the domain Safety.

### 3.2. Frequency Distribution of NIC Domains and Classes

The overall distribution of mapped interventions is shown in Table 2. Among the seven NIC domains, Physiological: Complex was the most frequently identified (n = 116, 56.9%), followed by Safety (n = 44, 21.6%) and Physiological: Basic (n = 25, 12.3%). At the class level, Respiratory Management (18.6%), Skin/Wound Management (15.7%), and Tissue Perfusion Management (14.2%) were the most frequently occurring classes, reflecting the high acuity of patient care needs in the target clinical environments. Notably, there were no clinical questions that corresponded to NIC domains of Family or Community, indicating that evidence needs in this study were concentrated within domains directly related to individual patient care in acute settings.

### 3.3. Most Frequently Identified Nursing Interventions Mapped to the NIC Taxonomy

Table 3 details the 20 most frequently cited NIC interventions. The highest-ranked intervention was Infection Control (6540) (n = 30, 14.7%), followed by Pressure Injury Prevention (3540) (n = 23), Airway Suctioning (3160) (n = 16), and Intravenous (IV) Insertion (4190) (n = 13). These top 20 interventions collectively accounted for 80% of all mapped responses, demonstrating a concentrated distribution of perceived evidence needs across a relatively narrow scope of nursing activities.

### 3.4. Visualization of Evidence Priorities

Figure 1, Figure 2 and Figure 3 display the Pareto charts that visualize the cumulative distribution of evidence needs across NIC domains, classes, and interventions. The charts demonstrate that a relatively small number of items in each category account for the majority of total frequencies, illustrating a classic Pareto distribution. Specifically, NIC interventions such as Infection Control and Pressure Injury Prevention, as well as classes like Respiratory Management and Skin/Wound Management, showed the highest concentration. Similarly, the domain of Physiological: Complex dominated overall clinical inquiry. This distribution underscores the importance of prioritizing a focused set of high-impact clinical topics when designing educational interventions and quality improvement strategies in EBNP.

## 4. Discussion

### 4.1. Mapping Clinical Questions Using Standardized Nursing Taxonomy

This study systematically mapped 204 clinical questions generated by frontline nurses in ICUs and EDs to the NIC taxonomy. Employing a structured methodology, practice-derived inquiries were translated into standardized terminologies, thereby facilitating rigorous analysis and integration into evidence-based nursing education, clinical guideline development, and quality improvement initiatives.

Guided by the Iowa Model of EBNP [10], the process of clinical question generation was affirmed as a critical entry point in the EBNP cycle. The use of both PICO-structured and narrative formats enabled comprehensive articulation of clinical concerns, reflecting the complexity and heterogeneity of real-world nursing practice. Mapping these questions to the NIC taxonomy allowed for quantification and categorization by domain, class, and intervention, thus supporting systematic prioritization of clinical needs [13]. These findings underscore the pivotal role of empowering bedside nurses to systematically express and structure their clinical inquiries. Such empowerment not only initiates a virtuous cycle of evidence generation and utilization but also strengthens the linkage between frontline practice and the advancement of nursing science [15,24]. The integration of standardized taxonomies, as demonstrated, provides a scalable and replicable framework for aligning clinical inquiry with the ongoing development of evidence-based nursing interventions [24].

### 4.2. Prioritization of Complex Physiological Interventions

In this study, the Physiological: Complex domain of the NIC taxonomy encompassed 56.9% of all mapped clinical questions, with Respiratory Management (18.6%) and Tissue Perfusion Management (14.2%) identified as the most frequently referenced classes. The interventions with the highest frequency—Infection Control (6540); Pressure Injury Prevention (3540); and Airway Suctioning (3160)—collectively accounted for over 40% of responses. These findings are consistent with previous research in ICU and ED settings, which underscore the critical need for robust, evidence-based guidance to inform clinical decision-making in the stabilization of physiological parameters, infection prevention, and ventilatory support [25,26,27]. The recurrent prioritization of these interventions across international studies highlights the persistent demand for targeted evidence synthesis and implementation, particularly in high-acuity environments where timely and effective nursing actions directly impact patient outcomes [15,22]. The convergence of these results with global literature reinforces the imperative for ongoing research and the development of best-practice protocols in managing acute physiological instability [28]. By elucidating the predominant focus areas within the Physiological: Complex domain, this study contributes to the advancement of EBNP and supports the optimization of care for critically ill patients [16].

### 4.3. Elevated Demand for Evidence in Patient Safety and Respiratory Care

In this study, 21.6% of the mapped clinical questions were classified within the safety domain, highlighting a pronounced demand among nurses for robust, structured evidence to inform patient safety practices, risk mitigation strategies, and emergency response protocols. Within this domain, inquiries related to mechanical ventilation constituted a critical subcategory, reflecting the complexities inherent in ventilator management in acute care settings. Existing literature consistently reports that nurses in ICUs and EDs frequently experience low self-efficacy in key aspects of ventilator care, including adjustment of ventilator parameters, alarm interpretation, artificial airway maintenance, and initiation of weaning protocols [25,26,29,30,31,32]. These gaps in knowledge and confidence are particularly concerning, as timely recognition and proactive management of ventilated patients are directly associated with improved clinical outcomes [33]. Evidence-based interventions—such as appropriate alarm setting; humidification; and meticulous oral care—have been demonstrated to significantly reduce the incidence of ventilator-associated pneumonia and related complications [29,34].

Moreover, international accreditation standards, including those established by the Joint Commission International (JCI), position patient safety as a fundamental metric of healthcare quality [35]. The study cohort, comprised of nurses from tertiary care institutions undergoing regular accreditation processes, exhibited heightened awareness of safety priorities, including fall prevention, medication error reduction, and the prevention of nosocomial infections [36]. This context likely contributed to the elevated frequency of safety-related clinical inquiries observed in the data.

Collectively, these findings underscore the urgent need for the development and dissemination of accessible, evidence-based, and scenario-driven guidelines addressing both ventilator care and broader patient safety protocols. The integration of high-impact educational content into competency-based training modules, standardized checklists, and clinical decision support systems is essential to bridge the persistent evidence-practice gap and to enhance the quality and safety of care delivered to critically ill patients [37]. Consistent with these domain-specific patterns, no clinical questions in this study were mapped to the Family or Community domains of NIC. This distribution likely reflects the immediate clinical priorities of high-acuity ICU and ED nursing practice, where urgent physiological concerns dominate frontline decision-making, rather than indicating that these domains are irrelevant or unimportant. Given the increasing emphasis in contemporary nursing on family engagement, transitional care, and hospital-to-community continuity [38,39,40], future research should explore evidence needs in these domains and evaluate their implications for post-acute care quality and long-term patient and family outcomes.

### 4.4. Strengthening Infrastructure for Evidence Translation and Access

Despite growing advocacy for EBNP, its consistent implementation within high-acuity care settings remains suboptimal [6,7,9]. A fundamental prerequisite for effective EBNP integration is the regular evaluation of departmental clinical protocols to ensure they are current, readily accessible, and reflective of the most recent evidence [1,4]. This process should encompass systematic review of guideline formats, update frequencies, and end-user usability to facilitate efficient knowledge application at the point of care [11,33]. In addition, the design of continuing education programs—particularly within critical care environments—should be informed by empirically identified evidence needs; as revealed through clinical question mapping. Educational priorities such as ventilator management, hemodynamic monitoring, and infection control ought to be determined by documented inquiry trends rather than anecdotal or administratively driven topics. Pragmatic strategies to enhance evidence translation include the development of concise, one- to two-page evidence summaries (e.g., “EBNP leaflets”) for high-frequency interventions and the integration of key recommendations directly into electronic nursing record (ENR) systems. Furthermore, retrospective analysis of nursing EMR data—specifically nursing diagnoses; interventions; and outcomes—can illuminate practice patterns and inform targeted evidence dissemination. Aligning frequently documented NNN linkages with prioritized knowledge translation efforts will further support the systematic advancement of EBNP in high-acuity settings [12,16].

### 4.5. Standardization Through NANDA–NIC–NOC Integration

The integration of the NNN framework in this study facilitated systematic and interpretable mapping of free-text clinical questions to standardized nursing intervention codes. This alignment enhances semantic interoperability and supports consistent data capture across diverse clinical settings, thereby advancing the development of evidence-based curricula and standardized documentation practices [12,13,15,16]. The structured application of the NNN framework is particularly advantageous in high-acuity environments, such as critical care units, where rapid clinical decision-making, effective interdisciplinary communication, and documentation efficiency are paramount [12]. By unifying nursing terminology across education, research, and clinical practice, the NNN framework provides a robust infrastructure for the scalable and sustainable implementation of EBNP.

### 4.6. Limitations and Future Research Directions

This study has several limitations that should be considered when interpreting the findings. It was conducted in a single tertiary-care hospital located in Gwangju, South Korea, and focused exclusively on ICUs and EDs. Therefore, the generalizability of the findings may be limited, as institutional characteristics, regional clinical protocols, and organizational cultures can differ markedly across healthcare institutions. Although the entire NIC taxonomy was used as the reference framework for cross-mapping, the clinical questions elicited from ICU and ED nurses resulted in a distribution of mapped interventions concentrated primarily in physiological health-related domains, with minimal representation of family- and community-focused domains. This distributional imbalance reflects the specialized focus and immediate priorities of high-acuity nursing practice, rather than any omission or lack of relevance of other NIC domains.

Future research should employ multicenter study designs encompassing diverse geographic regions and clinical settings, including general wards, outpatient clinics, and community-based facilities, to improve external validity and ensure broader applicability of the findings. Additionally, subgroup analyses based on nursing experience, clinical roles, and specialty areas are recommended to better capture the variability in the generation of clinical questions and associated evidence needs. Beyond study design considerations, further research is needed on the application of artificial intelligence (AI) and natural language processing (NLP) technologies to facilitate the automated mapping of clinical questions to standardized nursing taxonomies, such as the NIC [24]. Finally, longitudinal studies are recommended to evaluate the effectiveness of evidence-based nursing interventions, particularly in domains such as mechanical ventilation, early warning systems, and family-centered discharge planning. These investigations may provide critical insights to inform best practices and optimize patient outcomes in high-acuity care settings.

## 5. Conclusions

This study employed a structured cross-mapping approach, grounded in the Iowa Model of Evidence-Based Practice and the NANDA–NIC–NOC taxonomy, to systematically identify and prioritize clinical questions raised by frontline nurses in high-acuity care settings. The analysis revealed that most evidence needs were concentrated within the Physiological: Complex domain—particularly in relation to infection control; ventilator care; and tissue perfusion—followed by a strong emphasis on patient safety. Notably, the absence of inquiries related to family and community domains underscores persistent challenges in operationalizing holistic, transitional, and family-centered care within time-sensitive and procedure-intensive environments.

By standardizing the articulation of clinical questions into codified, actionable knowledge, this study provides a scalable and replicable framework for aligning frontline inquiry with evidence-informed education and clinical quality improvement. To bridge the evidence-practice gap in high-acuity settings, strategic investments are warranted in the development of tailored clinical guidelines, competency-based educational interventions, and ENR-integrated decision support tools.

Future research should adopt multicenter and interdisciplinary designs to improve generalizability, leverage emerging technologies such as artificial intelligence and natural language processing to enhance automated clinical mapping, and evaluate the longitudinal impact of targeted evidence-based interventions on patient safety and health outcomes. Ultimately, empowering nurses to systematically question, organize, and apply evidence remains foundational to advancing EBNP and achieving sustainable improvements in critical care delivery.

## Figures and Tables

**Figure 1 healthcare-13-01892-f001:**
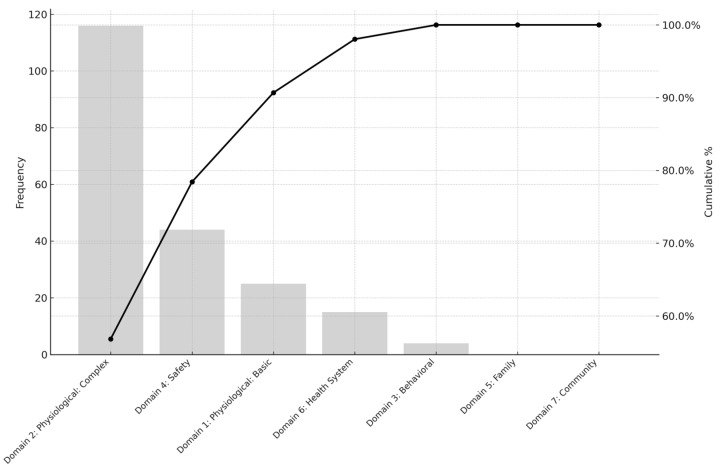
Pareto Chart of Cumulative Evidence Needs Across NIC Domains.

**Figure 2 healthcare-13-01892-f002:**
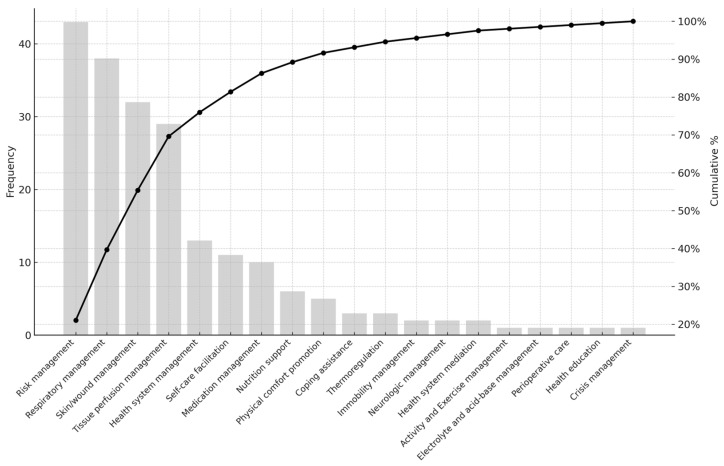
Pareto Chart of Cumulative Evidence Needs Across NIC Classes.

**Figure 3 healthcare-13-01892-f003:**
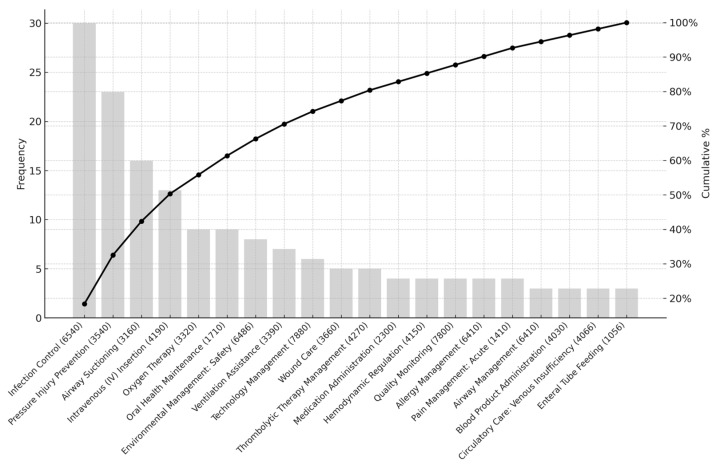
Pareto Chart of Cumulative Evidence Needs Across NIC Interventions.

**Table 1 healthcare-13-01892-t001:** Examples of mapping clinical questions to NIC interventions.

Clinical Question	Identified Nursing Action	NIC Interventions (Codes)	NIC Class	NIC Domain
“In patients requiring supplemental oxygen, does administration via nasal cannula at 5 L/min, compared to oxygen mask at the same flow rate, result in different clinical outcomes such as oxygen saturation, comfort, or respiratory effectiveness?”	Assess and monitor the effectiveness of oxygen therapy delivered via different modalities (nasal cannula vs. oxygen mask) in patients requiring supplemental oxygen.	Oxygen therapy (3320)	K. Respiratory management	2. Physiological: Complex
“In antibiotic skin testing (AST), what is the evidence supporting the standard practice of evaluating skin reactions 15 min after test administration?”	Assess and document hypersensitivity skin reactions at standardized intervals following antibiotic sensitivity testing.	Allergy management (6410)	V. Risk management	4. Safety
“In patients with a tracheostomy (artificial airway), is instilling a small amount of saline helpful for secretion clearance, and what are the associated side effects?”	Assess the therapeutic benefit and risks of saline instillation during suctioning	Airway suctioning (3160)	K. Respiratory management	2. Physiological: Complex
“In patients undergoing contrast-enhanced CT, is it necessary to secure a large-bore intravenous (IV) line (i.e., 18-gauge or smaller number) prior to the procedure to ensure adequate contrast administration?”	Secure and evaluate the adequacy of intravenous access for contrast administration in patients undergoing CT imaging.	Intravenous (IV) insertion (4190)	N. Tissue perfusion management	2. Physiological: Complex
“To facilitate evidence-based practice, what is the recommended minimum nurse-to-patient ratio for implementing team nursing in clinical settings?”	Establish appropriate nurse staffing levels to support team nursing and promote evidence-based practice in clinical settings.	Staff supervision (7830)	a. Health system management	6. Health management

Note: Clinical questions were anonymized and translated from Korean to English for reporting purposes.

**Table 2 healthcare-13-01892-t002:** Frequency Distribution of NIC Domains and Classes.

NIC Domain	N (%)	NIC Classes	N (%)
1. Physiological: Basic	25 (12.3)	A. Activity and Exercise management	1 (0.5)
		B. Elimination management	0 (0.0)
		C. Immobility management	2 (1.0)
		D. Nutrition support	6 (2.9)
		E. Physical comfort promotion	5 (2.5)
		F. Self-care facilitation	11 (5.4)
2. Physiological: Complex	116 (56.9)	G. Electrolyte and acid-base management	1 (0.5)
		H. Medication management	10 (4.9)
		I. Neurologic management	2 (1.0)
		J. Perioperative care	1 (0.5)
		K. Respiratory management	38 (18.6)
		L. Skin/wound management	32 (15.7)
		M. Thermoregulation	3 (1.5)
		N. Tissue perfusion management	29 (14.2)
3. Behavioral	4 (2.0)	O. Behavior therapy	0 (0.0)
		P. Cognitive therapy	0 (0.0)
		Q. Communication enhancement	0 (0.0)
		R. Coping assistance	3 (1.5)
		S. Health education	1 (0.5)
		T. Psychological comfort promotion	0 (0.0)
4. Safety	44 (21.6)	U. Crisis management	1 (0.5)
		V. Risk management	43 (21.1)
5. Family	0 (0)	W. Childbearing care	0 (0.0)
		Z. Childrearing care	0 (0.0)
		X. Lifespan care	0 (0.0)
6. Health system	15 (7.4)	Y. Health system mediation	2 (1.0)
		a. Health system management	13 (6.4)
		b. Information management	0 (0.0)
7. Community	0 (0)	c. Community health promotion	0 (0.0)
		d. Community risk management	0 (0.0)

**Table 3 healthcare-13-01892-t003:** Most Frequently Identified Nursing Interventions Mapped to the NIC Taxonomy.

Rank	Domain	Classes	Interventions (Codes)	Frequency (n)	Cumulative %
1	4. Safety	V. Risk management	Infection control (6540)	30	14.7
2	2. Physiological: Complex	L. Skin/Wound management	Pressure injury prevention (3540)	23	26.0
3	2. Physiological: Complex	K. Respiratory management	Airway suctioning (3160)	16	33.8
4	2. Physiological: Complex	H. Medication management	Intravenous (IV) insertion (4190)	13	40.2
5	2. Physiological: Complex	K. Respiratory management	Oxygen therapy (3320)	9	44.6
6	1. Physiological: Basic	F. Self-care facilitation	Oral health maintenance (1710)	9	49.0
7	4. Safety	V. Risk management	Environmental management: safety (6486)	8	53.0
8	2. Physiological: Complex	K. Respiratory management	Ventilation assistance (3390)	7	56.4
9	6. Health system	a. Health system management	Technology management (7880)	6	59.3
10	2. Physiological: Complex	L. Skin/Wound management	Wound care (3660)	5	61.8
11	2. Physiological: Complex	N. Tissue/Perfusion management	Thrombolytic therapy management (4270)	5	64.2
12	2. Physiological: Complex	H. Medication management	Medication administration (2300)	4	66.2
13	2. Physiological: Complex	N. Tissue/Perfusion management	Hemodynamic regulation (4150)	4	68.1
14	6. Health system	a. Health system management	Quality monitoring (7800)	4	70.1
15	4. Safety	V. Risk management	Allergy management (6410)	4	72.1
16	1. Physiological: Basic	E. Physical comfort promotion	Pain management: acute (1410)	4	74.0
17	2. Physiological: Complex	K. Respiratory management	Airway management (6410)	3	75.5
18	2. Physiological: Complex	N. Tissue/Perfusion management	Blood product administration (4030)	3	77.0
19	2. Physiological: Complex	N. Tissue/Perfusion management	Circulatory care: venous insufficiency (4066)	3	78.4
20	1. Physiological: Basic	D. Nutritional support	Enteral tube feeding (1056)	3	80.0

## Data Availability

The raw data supporting the conclusions of this article will be made available by the author on request.

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
