# Peer review of "Mapping Clinical Questions to the Nursing Interventions Classification: An Evidence-Based Needs Assessment in Emergency and Intensive Care Nursing Practice in South Korea"

_healthcare, 2025, doi:10.3390/healthcare13151892_

Round 1

Reviewer 1 Report

Comments and Suggestions for Authors

The article falls within the thematic scope of the journal. Nevertheless, it requires some clarifications, corrections, or modifications, as detailed below:

The study was conducted in South Korea. I believe that the title of the article should include the name of the country, as the current version may mislead the reader. The reader might assume that the study was conducted in multiple countries. Therefore, the phrase “in South Korea” should be added to the title. This is a methodological requirement.

Line 54: The author mentions the use of the IOWA Model. However, there is no explanation as to why other models were not considered. Was the choice of this model arbitrary, or is it supported by specific arguments?

Line 104: When recruiting respondents for the study, the author used the following criterion: “a minimum of six months’ experience in the current unit.” This raises the question: why was the six-month threshold chosen? Was this criterion selected arbitrarily, or was it based on any guidelines? This requires clarification.

Line 344: In the section “Limitations and Future Research Directions,” it should be clearly stated that the study was conducted in South Korea. It is not sufficient to merely mention that the study was carried out in a single facility (hospital). Furthermore, since the study only involves one facility, it is unclear whether the results can be considered representative of other hospitals in South Korea. To make such a claim, further studies would need to be conducted in other facilities and compared with each other.

Reviewer 2 Report

Comments and Suggestions for Authors

I am very honored to be invited to participate in the judging. This manuscript explores the mapping of clinical problems to the classification of nursing interventions through an analysis of emergency and critical care practice situations. It has very good practical and theoretical significance. However, some problems are still worth exploring and improving.

1. The author needs to effectively define "nurses must make fast, high-risk clinical decisions" in the introduction. More often than not, at least in my perception, critical clinical decisions should be made by doctors, not nurses. The author must answer this question carefully and clearly distinguish which decisions have a significant impact on patients are made by nurses. This issue should be articulated clearly, given the possible differences in clinical regimes in different countries.

2 It is worth noting that this paper uses NIC version 8. But at the same time the title of this paper targets the ICU and EU scenarios. So is everything covered in NIC or is it covered in part? The author should express himself more clearly.

3 Data analysis did not appear to be particularly problematic.

4 The figures 1, 2, and 3 mentioned in Section 3.4 do not appear in the article.

5. The conclusions need to be reexamined. As this manuscript focuses on ICU and ED care, home and community-oriented care does not seem to be applicable here. I think authors should focus on current issues and not expand on them.

Round 2

Reviewer 1 Report

Comments and Suggestions for Authors

I have no more comments.